# Performance of modeling and balancing approach methods when using weights to estimate treatment effects in observational time-to-event settings

Guilherme W. F. Barros<sup></sup>*, Marie Eriksson, Jenny Häggström

Department of Statistics, Umeå School of Business, Economics and Statistics, Umeå University, Umeå, Sweden

* guilherme.barros@umu.se

## Abstract

In observational studies weighting techniques are often used to overcome bias due to confounding. Modeling approaches, such as inverse propensity score weighting, are popular, but often rely on the correct specification of a parametric model wherein neither balance nor stability are targeted. More recently, balancing approach methods that directly target covariate imbalances have been proposed, and these allow the researcher to explicitly set the desired balance constraints. In this study, we evaluate the finite sample properties of different modeling and balancing approach methods, when estimating the marginal hazard ratio, through Monte Carlo simulations. The use of the different methods is also illustrated by analyzing data from the Swedish stroke register to estimate the effect of prescribing oral anticoagulants on time to recurrent stroke or death in stroke patients with atrial fibrillation. In simulated scenarios with good overlap and low or no model misspecification the balancing approach methods performed similarly to the modeling approach methods. In scenarios with bad overlap and model misspecification, the modeling approach method incorporating variable selection performed better than the other methods. The results indicate that it is valuable to use methods that target covariate balance when estimating marginal hazard ratios, but this does not in itself guarantee good performance in situations with, e.g., poor overlap, high censoring, or misspecified models/balance constraints.

## Introduction

Establishing causal relationships is a primary objective of scientific research, and randomized controlled trials (RCTs), which allow for unbiased estimation of average treatment effects [1], are often regarded as the gold standard. However, RCTs involving human subjects have practical limitations relating to, e.g., study duration and ethics, and there is a growing interest in using observational data to emulate the conditions found in RCTs, e.g., by removing confounding bias, and establishing causal links between variables [2, 3].

**Data Availability Statement:** The code used to generate the data in the Monte Carlo simulation is available on the github repository: https://github.com/Wangbarros/Modeling_vs_Balancing_Time_

to_Event. The data analyzed in the case study consist of third party data from Riksstroke and according to Swedish legislation (https://etikprovningsmyndigheten.se/for-forskare/vadsager-lagen/) data cannot be made available for use beyond what has been approved by the ethical review authority. Therefore, the data cannot be made publicly available. Data may be made available from Riksstroke (contact via riksstroke@regionvasterbotten.se) upon reasonable request by researchers who meet the criteria for access to confidential data according to Swedish laws and regulations. The data underlying the results presented in the case study are available from Rikstroke, the swedish stroke register. The data on which the simulation results are based is made available through the simulation study code.

**Funding:** This work was supported by the Swedish Research Council (Dnr: 2018–01610). The funders had no role in study design, data collection and analysis, decision to publish, or preparation of the manuscript.

**Competing interests:** The authors have declared that no competing interests exist.

Weighting methods are commonly used for this purpose, wherein weights are used to adjust and balance the empirical distributions of the observed covariates with the goal of making the treatment groups similar in terms of background characteristics. Traditionally, weights are calculated using the propensity score (PS), i.e., the probability of receiving treatment conditional on the observed covariates, and the use of inverse probability of treatment weighting (IPTW) [4] is widespread. PS is typically estimated using a regression model such as logistic regression, and it is known that resulting weights can exhibit high variability, leading to instability of final effect estimates [5, 6]. This approach to weighting has been called the 'modeling approach' [7], since focus is on maximizing the fit of a treatment assignment model which is later used to derive weights. A more recent approach to weighting is the balancing approach [7], which encompasses methods that directly find weights with certain features, without explicitly specifying a functional form for the underlying PS model. The use of IPTW to estimate treatment effects, similar to those reported in RCTs, in a survival outcome setting has been described in previous work by Austin [8], and evaluated in a series of Monte Carlo simulations [9]. However, to our knowledge, the finite sample properties of balancing approach methods have not been studied in this context.

The aim of this study was to investigate the finite sample properties of different weighting methods when used to estimate population level treatment effects, i.e., marginal treatment effects, using survival data. The paper is structured as follows: First, relevant causal inference and survival analysis concepts are reviewed; second, methods within the modeling and balancing approaches to weighting are described; third, the design and results of an extensive simulation study are described; fourth, the weighting methods are applied to a real dataset; and last, the results are discussed.

## Materials and methods

### Ethical considerations

Statistical method development for fair comparisons of stroke care and outcome was part of the EqualStroke-project, approved by the Ethical Review Board in Umeå (Dnr: 2012–321-31M, 2014–76-32M). Patients and next of kin are informed about the registration and aim of the Riksstroke-register and their right to decline participation (opt-out consent).

### Treatment effects and survival analysis

In the potential outcomes framework [10, 11], every subject $i$ is associated with a vector of observed baseline covariates $\mathbf{X}_i$, treatment status $Z_i$ and pair of potential outcomes $Y_i(0)$ and $Y_i(1)$. These latter denote, respectively, the outcome under no treatment ($Z_i = 0$) and outcome under (active) treatment ($Z_i = 1$). The observed outcome is $Y_i = Z_i Y_i(1) + (1 - Z_i) Y_i(0)$. Two common parameters of interest are the average treatment effect, $ATE = E[Y_i(1) - Y_i(0)]$, and the average treatment effect on the treated, $ATT = E[Y_i(1) - Y_i(0)|Z = 1]$. Under randomization $ATE = ATT$, but this does not hold in observational studies due to the existence of confounding variables. With a nonrandomized treatment assignment, ATE and ATT can only be identified under the assumptions of unconfoundedness ($Y(0), Y(1) \perp\!\!\!\perp Z|\mathbf{X}$), overlap ($0 < Pr(Z = 0|\mathbf{X}) < 1$), and no potential outcome of a subject being affected by the assignment of treatments to the other subjects (stable unit treatment assumption; SUTVA).

In a survival analysis setting, the potential outcomes of a subject are the time to some event of interest under treatment, and no treatment. $\{T_i, D_i, Z_i, \mathbf{X}_i; i = 1, 2, \ldots, n\}$ denotes independent and identically distributed data for $n$ subjects, where $C_i$ is the censoring time, $D_i = 1_{Y_i \leq C_i}$ the event indicator, $T_i = \min\{Y_i, C_i\}$ the observed time, and $Y_i, Z_i$ and $\mathbf{X}_i$ are defined as above.

$n_1 = \sum_{i=1}^{n} Z_i$ and $n_0 = n - n_1$ denotes the sample size in the treatment and no treatment group, respectively. The censoring is assumed to be independent, i.e., $C_i \perp\!\!\!\perp Y_i | Z_i, \mathbf{X}_i$. In this right-censored setting ATE and ATT, defined as mean differences in survival time, may not be possible to estimate nonparametrically [12].

Arguably, the most common estimand in survival settings is the conditional hazard ratio (CHR), which is typically estimated by a Cox model [13], where a hazard at time $t$ is given by

$$\lambda(t|Z, \mathbf{X}) = \exp(\alpha_Z Z + \mathbf{X}^\top \boldsymbol{\alpha})\lambda_0(t), \tag{1}$$

where $\lambda_0$ is the baseline hazard function, $\alpha_Z$ is the parameter relating the treatment variable to the hazard, and $\boldsymbol{\alpha}$ is a column vector of parameters relating the covariates to the hazard. The CHR is given by $\exp(\alpha_Z)$. A hazard ratio that is not conditional on a certain set of covariate values is the marginal hazard ratio (MHR), roughly the hazard ratio we would see when applying the treatment to an entire population [14]. Given the Cox model in 1, CHR is equal to MHR in randomized studies, which is not expected in observational studies. If the treatment assignment is randomized, MHR can be estimated using a Cox model that includes only the treatment variable $Z$. In an observational study, weights that adjust for differences in $\mathbf{X}$ are required in addition to the latter model [8, 12]. Similar to ATE and ATT, we can be interested in MHR in an entire population or MHR in the treated population. We define $MHR_{ATE}$ as the MHR that is obtained if a dataset containing both the potential outcomes of all individuals is used to fit a Cox model with the treatment status indicator as the sole covariate. Similarly, we define $MHR_{ATT}$ as the MHR obtained if this hypothetical analysis was restricted to those that actually received the treatment.

In this study we focus on estimating MHR, since it is a measure often reported in RCTs. Note however that, even in RCTs, MHR does not have a proper causal interpretation, because the initial unconfoundedness at baseline is broken after the first failure event [15, 16]. For recently proposed alternative estimands see Mao [12] and references therein.

## Weighting methods

Since the treatment assignment is nonrandomized in observational studies, it is necessary to balance the covariates of the data to facilitate unbiased estimation of treatment effects. The data can be considered balanced with respect to $\mathbf{X}$ if the probability distribution of $\mathbf{X}$ is similar in the treated and untreated groups, i.e., $Pr(\mathbf{X}|Z = 1) = Pr(\mathbf{X}|Z = 0)$ [17]. Weighting methods are commonly used to balance covariates due to not requiring modeling of the outcome [18].

**Balance assessment.**   Balance can be assessed using a variety of methods [19]; the most common, and the one employed in this paper, is comparing means and proportions of covariates between treated and untreated subjects [20, 21]. For a continuous covariate $X$, we define the absolute standardized mean difference as

$$d = \frac{|\bar{X}_{\text{treated}} - \bar{X}_{\text{untreated}}|}{\sqrt{(s^2_{\text{treated}} + s^2_{\text{untreated}})/2}},$$

where $\bar{X}_{\text{treated}}$ and $\bar{X}_{\text{untreated}}$ are the sample means of $X$ in treated and untreated subjects, respectively, and $s^2_{\text{treated}}$ and $s^2_{\text{untreated}}$ the analogous sample variances. For binary covariates, $d$ is defined as the absolute unstandardized difference in proportions, since these are already on the same scale. The higher $d$, the more disparate are the means of the two populations. A covariate is generally considered to be 'balanced' if $d$ is less than 0.25, but stricter thresholds, e.g., 0.10, have also been suggested [20, 22, 23].

**Modeling approach methods.**   The most common weighting method for balancing covariates uses PS, $Pr(Z = 1|\mathbf{X})$, since if the assumption of unconfoundedness is true given the

observed covariates, it also holds given PS [18]. Using the inverse of PS as a weighting factor, it is possible to estimate treatment effects on a population level. With $\gamma = 1$ if the target parameter is $MHR_{ATE}$ and $\gamma = 0$ if it is $MHR_{ATT}$, the IPTW weights are defined as

$$w_i = \left[ \frac{\gamma}{\Pr(Z_i = 1|\mathbf{X}_i)} + (1 - \gamma) \right] \left[ Z_i + \frac{\Pr(Z_i = 1|\mathbf{X}_i)(1 - Z_i)}{1 - \Pr(Z_i = 1|\mathbf{X}_i)} \right].$$

For $MHR_{ATE}$, IPTW gives larger weights to treated subjects with low PS and untreated subjects with high PS. The main goal of the weights is to balance the covariates, making it possible to estimate the parameter of interest without bias. However, in some cases the estimated weights have high variance themselves, due to some subjects having very high or very low PS, which then produces parameter estimates with high variance. With a correctly specified PS model the highly variable weights accurately describe reality, but in practice we do not know if the model was in fact correctly specified.

In practice PS has to be estimated and, although several machine learning methods can be applied, the most commonly used method is logistic regression, the advantages of which are simplicity, ease of implementation and interpretation, and familiarity to researchers in a variety of disciplines [24]. Henceforth, IPTW with logistic regression is referred to in this paper as 'GLM'. In situations where the covariate vector (potentially including transformations, e.g., higher order terms and interactions) is of high dimensionality, least absolute shrinkage and selection operator (LASSO) regularization [25] is an option. Fitting a prespecified PS model using logistic regression or using LASSO regularization with a shrinkage parameter selected by cross-validation (which is common practice) implies targeting treatment assignment prediction, rather than covariate balance and treatment effect estimation [26]. However, LASSO regularization with a shrinkage parameter selection strategy directly targeting the balance of the covariates is possible [27]. In this study the shrinkage parameter is selected such that the average balance of the covariates is maximized, i.e., the average $d$ after weighting is minimized (we will refer to this as LASSO). It should be noted that the outcome itself is never used in this selection process.

**Balancing approach methods.** In contrast to PS methods, which often rely on specification of a parametric regression model, methods without the need to specify a functional form for the PS, such as the class of minimal weights [28], have been developed. If the target parameter is $MHR_{ATT}$, the aim is to find weights to reweight the untreated subjects such that the reweighted untreated sample has similar covariate distributions as the treated sample. For this purpose, the class of minimal weights, explicitly targeting both covariate balance and stability of the weights, is solved for the following mathematical program [7, 28]:

$$\begin{aligned} \underset{w}{\text{minimize}} \quad & \sum_{i=1}^{n} (1 - Z_i) f(w_i) \\ \text{subject to} \quad & \left| \sum_{i=1}^{n} w_i (1 - Z_i) B_p(X_i) - \frac{1}{n_1} \sum_{i=1}^{n} Z_i B_p(X_i) \right| \le \delta_p, \ p = 1, \ldots, P, \end{aligned}$$

(2)

where $f$ is a convex function of the weights $w$, and $B_p(X_p)$ are smooth functions of the covariates. It is advisable to balance not only the original covariates but also transformations, e.g., basis functions of the covariates. Hence, the term 'covariate' when used in relation to 2 above, can also refer to a such a transformation. $\delta_p$ are tolerance values that limit the distance between the weighted mean and the mean of the covariates. In addition to the balance constraints in 2,

the weights can also be subject to normalizing constraints:

$$\sum_{i=1}^{n} (1 - Z_i) w_i = 1 \quad \text{and} \quad w_i \geq 0.$$

It should be also noted that, for the estimation of ATT weights, all treated individuals receive the same weights, and the sum of the weights for the treated individuals will also be equal to 1. Special cases of minimal weights are: entropy balancing (EB) weights [29] with $f(x) = x \log(x/q)$ (usually $q = 1/n_0$) and $\delta = 0$; the empirical balancing calibration (CAL) weights [30] with $f(x) = D(x, 1)$ and $\delta = 0$, where $D(x, x_0)$ is a distance measure for a fixed $x_0 \in \mathbb{R}$ (continuously differentiable in $x_0$, nonnegative and strictly convex in $x$); and stable balancing weights (SBW) [31] with $f(x) = (x - 1/n_0)^2$ and $\delta \in \mathbb{R}_0^+$. Since $\delta = 0$ for EB and CAL these methods result in exact balancing, while SBW results in approximate balancing. Under certain assumptions, minimal weights, e.g., SBW, consistently estimate the true inverse PS weights [28], albeit in a different way than traditional modeling approaches to weighting [32].

When the covariates are balanced in an approximate manner, $\delta_p > 0$, it is in practice necessary for the researcher to either assign individual values for $\delta_p$, $p = 1, \ldots, P$, or choose a single tolerance factor for $\delta$ that is scaled for each covariate according to the covariate's standard deviation. In Algorithm 1 below an algorithm for selecting $\delta$ is presented [28, 33].

**Algorithm 1** Tuning $\delta$

```
for Each δ in a grid 𝔇 of possible imbalances (in units of standard
deviation) do
  Compute wᵢ(δ) by solving Eq 2 using the original dataset Sₒ
  for each b ∈ 1, ..., B do
    Take a bootstrap sample S_b from Sₒ
    Evaluate the covariate balance C_b(δ) on S_b
  end for
  Compute the mean covariate balance C(δ) = 1/B Σ_{b=1}^{B} C_b(δ)
end for
Output δ* = argmin_{δ∈𝔇} C(δ)
```

When the target parameter is $MHR_{ATT}$ the covariate balance measure is

$$C_b(\delta) = \frac{1}{P} \sum_{p=1}^{P} \left| \frac{\sum_{i \in S_b} (1 - Z_i) w_i(\delta) B_p(X_i)}{\sum_{i \in S_b} (1 - Z_i) w_i(\delta)} - \frac{\sum_{i \in S_o} Z_i B_p(X_i)}{\sum_{i \in S_o} Z_i} \right| \frac{1}{s_{o,p}^0},$$

where

$$s_{o,p}^0 = \sqrt{\frac{\sum_{i \in S_o} (1 - Z_i)(B_p(X_i) - \hat{\mu}_{o,p}^0)^2}{(\sum_{i \in S_o} (1 - Z_i)) - 1}}$$

and

$$\hat{\mu}_{o,p}^0 = \frac{1}{\sum_{i \in S_o} (1 - Z_i)} \sum_{i \in S_o} (1 - Z_i) B_p(X_i)$$

is the standard deviation and mean of covariate $p$ in the untreated subsample of $S_o$, respectively. The algorithm is based on the idea that an optimal tolerance value for $\delta$ is one that balances not only the population but also any draws from the same population. As such, covariate balance is evaluated on bootstrapped samples considering the weights estimated in the original dataset. This does not guarantee that the selected value is optimal in a given problem, but it has

been shown that Algorithm 1 selects $\delta$ in a manner that is optimal or close to optimal, in terms of root mean squared error (RMSE) [28].

Estimating $MHR_{ATE}$ instead of $MHR_{ATT}$ requires solving an optimization problem, similar to the one presented in 2, twice. In the first step the weights needed to reweight the untreated sample, such that it has similar covariate distributions to the full sample, are ascertained; in the second step weights needed to reweight the treated sample such that it becomes similar to the full sample are found.

EB was initially proposed as a preprocessing method without consideration of how this would impact any subsequent inference [29], but has since been shown to be doubly robust in settings where the outcome model is linear and the PS model is logistic [34]. CAL and SBW have been shown to be semiparametrically efficient for estimating ATE in a non-survival setting [7, 28, 30]. Depending on how the objective function is formulated, CAL can be used to derive, e.g., exponential tilting weights (CAL-ET; equivalent to EB with uniform base weights) and empirical likelihood weights [30]. EB/CAL-ET and SBW have been shown to outperform GLM in some non-survival settings [29, 31, 35–40] and some survival settings [36, 39]. Comparisons between EB/CAL-ET and SBW when estimating ATE and ATT in a non-survival setting have shown that the methods performed similarly in 'good overlap' settings but SBW outperformed EB/CAL-ET in 'bad overlap' settings [28].

A related method, the Covariate Balancing Propensity Score (CBPS) was introduced as a method for simultaneously optimizing covariate balance and parametrically estimating PS [41]. Later, a nonparametric version of CBPS (npCBPS) was developed [42] wherein there is no need to specify a functional form for PS. npCBPS finds weights that maximize the empirical likelihood for certain balancing constraints. The npCBPS optimization procedure can, however, be slow and the problem does not always admit a solution, since the empirical likelihood is not generally convex.

## Monte Carlo simulation

Simulations were performed to study the finite sample properties of weighting methods when estimating $MHR_{ATT}$ for various data generating processes (DGPs). Each setup was iterated 1000 times. Data generation and all computations were performed with the software R [43].

**Generating data.** A framework that was suitable for studying estimation under model misspecification was used to generate data [9, 44]. Ten independent covariates, $X_1$ to $X_{10}$, were simulated; $X_2$, $X_4$, $X_7$, and $X_{10}$ were standard normally distributed and the other six were Bernoulli distributed with a success probability of 0.5. Observational data was then generated according to seven different scenarios (A to G), ranging from a linear additive treatment assignment model to more complex models exhibiting various degrees of nonlinearity and/or nonadditivity. $X_1$ to $X_4$ were directly related to the probability of receiving treatment and to the time-to-event outcome; $X_5$ to $X_7$ were directly associated only with the probability of treatment; and $X_8$ to $X_{10}$ were only directly connected to the time-to-event outcome. The subject-specific probability of treatment for each scenario was:

(A). Additivity and linearity (main effects only):

$$\text{logit}(\Pr(Z_i = 1|\mathbf{X}_i)) = \beta_0 + \beta_1 X_{1,i} + \beta_2 X_{2,i} + \beta_3 X_{3,i}\beta_4 X_{4,i} +$$

$$\beta_5 X_{5,i} + \beta_6 X_{6,i} + \beta_7 X_{7,i}$$

(B). Mild nonlinearity (one quadratic term):

$$\begin{aligned}\text{logit}(\Pr(Z_i = 1|\mathbf{X}_i)) = \ & \beta_0 + \beta_1 X_{1,i} + \beta_2 X_{2,i} + \beta_3 X_{3,i} + \beta_4 X_{4,i} + \\ & \beta_5 X_{5,i} + \beta_6 X_{6,i} + \beta_7 X_{7,i} + \beta_2 X_{2,i}^2.\end{aligned}$$

(C). Moderate nonlinearity (three quadratic terms):

$$\begin{aligned}\text{logit}(\Pr(Z_i = 1|\mathbf{X}_i)) = \ & \beta_0 + \beta_1 X_{1,i} + \beta_2 X_{2,i} + \beta_3 X_{3,i} + \beta_4 X_{4,i} + \\ & \beta_5 X_{5,i} + \beta_6 X_{6,i} + \beta_7 X_{7,i} + \beta_2 X_{2,i}^2 + \\ & \beta_4 X_{4,i}^2 + \beta_7 X_{7,i}^2.\end{aligned}$$

(D). Mild nonadditivity (four interaction terms):

$$\begin{aligned}\text{logit}(\Pr(Z_i = 1|\mathbf{X}_i)) = \ & \beta_0 + \beta_1 X_{1,i} + \beta_2 X_{2,i} + \beta_3 X_{3,i} + \beta_4 X_{4,i} + \\ & \beta_5 X_{5,i} + \beta_6 X_{6,i} + \beta_7 X_{7,i} + 0.5 \times \beta_1 X_{1,i} X_{3,i} + \\ & 0.7 \times \beta_2 X_{2,i} X_{4,i} + 0.5 \times \beta_4 X_{4,i} X_{5,i} + 0.5 \times \beta_5 X_{5,i} X_{6,i}.\end{aligned}$$

(E). Mild nonadditivity and nonlinearity (one quadratic term and four interaction terms):

$$\begin{aligned}\text{logit}(\Pr(Z_i = 1|\mathbf{X}_i)) = \ & \beta_0 + \beta_1 X_{1,i} + \beta_2 X_{2,i} + \beta_3 X_{3,i} + \beta_4 X_{4,i} + \beta_5 X_{5,i} + \\ & \beta_6 X_{6,i} + \beta_7 X_{7,i} + \beta_2 X_{2,i}^2 + 0.5 \times \beta_1 X_{1,i} X_{3,i} + \\ & 0.7 \times \beta_2 X_{2,i} X_{4,i} + 0.5 \times \beta_4 X_{4,i} X_{5,i} + 0.5 \times \beta_5 X_{5,i} X_{6,i}.\end{aligned}$$

(F). Moderate nonadditivity (ten interaction terms):

$$\begin{aligned}\text{logit}(\Pr(Z_i = 1|\mathbf{X}_i)) = \ & \beta_0 + \beta_1 X_{1,i} + \beta_2 X_{2,i} + \beta_3 X_{3,i} + \beta_4 X_{4,i} + \beta_5 X_{5,i} + \\ & \beta_6 X_{6,i} + \beta_7 X_{7,i} + 0.5 \times \beta_1 X_{1,i} X_{3,i} + 0.7 \times \beta_2 X_{2,i} X_{4,i} + \\ & 0.5 \times \beta_3 X_{3,i} X_{5,i} + 0.7 \times \beta_4 X_{4,i} X_{6,i} + 0.5 \times \beta_5 X_{5,i} X_{7,i} + \\ & 0.5 \times \beta_1 X_{1,i} X_{6,i} + 0.7 \times \beta_2 X_{2,i} X_{3,i} + 0.5 \times \beta_3 X_{3,i} X_{4,i} + \\ & 0.5 \times \beta_4 X_{4,i} X_{5,i} + 0.5 \times \beta_5 X_{5,i} X_{6,i}.\end{aligned}$$

(G). Moderate nonadditivity and nonlinearity (three quadratic terms and ten interaction terms):

$$\begin{aligned}\text{logit}(\Pr(Z_i = 1|\mathbf{X}_i)) = \ & \beta_0 + \beta_1 X_{1,i} + \beta_2 X_{2,i} + \beta_3 X_{3,i} + \beta_4 X_{4,i} + \beta_5 X_{5,i} + \\ & \beta_6 X_{6,i} + \beta_7 X_{7,i} + 0.5 \times \beta_1 X_{1,i} X_{3,i} + 0.7 \times \beta_2 X_{2,i} X_{4,i} + \\ & 0.5 \times \beta_3 X_{3,i} X_{5,i} + 0.7 \times \beta_4 X_{4,i} X_{6,i} + 0.5 \times \beta_5 X_{5,i} X_{7,i} + \\ & 0.5 \times \beta_1 X_{1,i} X_{6,i} + 0.7 \times \beta_2 X_{2,i} X_{3,i} + 0.5 \times \beta_3 X_{3,i} X_{4,i} + \\ & 0.5 \times \beta_4 X_{4,i} X_{5,i} + 0.5 \times \beta_5 X_{5,i} X_{6,i} + \beta_2 X_{2,i}^2 + \beta_4 X_{4,i}^2 + \beta_7 X_{7,i}^2.\end{aligned}$$

With $\boldsymbol{\beta}^T = k(0.00, 0.80, -0.25, 0.60, -0.40, -0.80, -0.50, 0.70)$, $k$ was used to control the overlap between the treated and untreated groups. $k = 1$ represented strong overlap between the two groups, and weakened as $k$ increased. For each subject, $Z_i$ was then drawn from a Bernoulli distribution. The time-to-event outcome was generated as

$$Y_i = \left( \frac{-\log(u_i)}{\lambda \exp(\text{LP}_i)} \right)^{1/\eta}$$

where $u_i \sim U(0, 1)$, $\lambda = 0.00002$, $\eta = 2$ (as in [8]) and the linear predictor $\text{LP}_i = \alpha_Z Z_i + \alpha_1 X_{1,i} + \alpha_2 X_{2,i} + \alpha_3 X_{3,i} + \alpha_4 X_{4,i} + \alpha_5 X_{8,i} + \alpha_6 X_{9,i} + \alpha_7 X_{10,i}$, with $\boldsymbol{\alpha} = (\alpha_Z, 0.30, -0.36, -0.73, -0.20, 0.71, -0.19, 0.26)$. This DGP resulted in data with $CHR_{ATT} = \exp(\alpha_Z)$. In order to generate data with a predetermined $MHR_{ATT}$, an iterative bisection method was used to determine the value of $CHR_{ATT}$ that resulted in the desired $MHR_{ATT}$ [14]. Data with independent censoring was generated as in Wan [45], see the S1 File for details.

**Statistical analyses in generated datasets.**   Weights were calculated using the methods described previously and $MHR_{ATT}$ was estimated by fitting a Cox model to the weighted sample, including the $Z$ as the only explanatory variable. For comparison, the same model was fit to the initial unweighted sample, and we refer to this strategy as NAIVE. Confidence intervals were based on standard errors, estimated using a robust sandwich-type variance estimator, taking the weighted nature of the data into account. Although no theoretical justification has been provided, simulation results have indicated that this estimator slightly overestimates the variance, and results in somewhat conservative confidence intervals when estimating $MHR_{ATE}$ and $MHR_{ATT}$ with IPTW [46]. However, more recently the possibility of anti-conservative inference when estimating ATT with IPTW was shown [47]. The R packages `glmnet` [48], `CBPS` [49], `ATE` [50], `sbw` [33], and `survival` [51] were used for LASSO, npCBPS, CAL-ET, SBW, and Cox modeling, respectively. For npCBPS and CAL-ET, the default options for the respective functions were used. For SBW, Gurobi [52] was used as solver and Algorithm 1 (implemented in `sbw`) was used to select $\delta$; otherwise, the default options were used.

**Experiments.**   To study how particular features of data and covariate sets affected the performance of the weighting methods, three different experiments, summarized in Table 1 were conducted. In all of the experiments, $MHR_{ATT} = 0.8$, $n = 1500$. As several of the methods were nonparametric, the focus was evaluating the performance of the weighting methods in data generation scenarios, which often demand flexibility as regards weight estimation. Therefore, scenarios with varying degrees of misspecification mixed with overlap and censoring were used, as these are important considerations for time-to-event data. In addition, a case in which there was no misspecification, analogous to one in which a researcher might have overspecified the model by adding unnecessary interactions and quadratic terms while trying to avoid misspecification, was investigated. Experiment 1 investigated how misspecification of PS models/balance constraints affected the estimation of $MHR_{ATT}$. Only the main effects of $\mathbf{X}_{small} = (X_1, X_2, X_3, X_4, X_5, X_6, X_7)^T$ were included in the PS models, and constraints were set only on $\mathbf{X}_{small}$ for the balancing approach methods. In addition, the degree of overlap ($k = 1, 2, 3$) was

**Table 1. The conducted experiments.** $k$ is the degree of overlap, $\pi$ is the censoring proportion, $\mathbf{X}_{small} = (X_1, \ldots, X_7)^T$ while $\mathbf{X}_{large}$ consists of $\mathbf{X}_{small}$ as well as all quadratic and two-way interaction terms computed from $\mathbf{X}_{small}$.

| Experiment | Data generation | | Estimation Covariates |
|---|---|---|---|
| | $k$ | $\pi$ | |
| 1: Misspecification and overlap | 1, 2, 3 | 0.0 | $\mathbf{X}_{small}$ |
| 2: Misspecification and censoring | 1 | 0.1, . . ., 0.5 | $\mathbf{X}_{small}$ |
| 3: Overspecification | 1 | 0.0 | $\mathbf{X}_{large}$ |

varied. For Scenario (A) the PS models/balance constraints are correctly specified, and can be used to compare the effect of just changing the overlap specifically. In Experiment 2, the rate of censoring ($\pi$ = 0.1, 0.2, 0.3, 0.4, 0.5) was varied rather than the degree of overlap. The same type of data was generated as in Experiment 1, albeit only for the overlap setting $k$ = 1. Researchers conducting data analysis generally try to avoid misspecification of the type exemplified here by including covariate transformations. If more terms are used for estimation than are necessary the model is overspecified, which may affect its performance. In Experiment 3 the effect of overspecification of PS models/balance constraints on estimation of $MHR_{ATT}$ was investigated by including both the main effects and quadratic and two-way interaction terms between covariates in $\mathbf{X}_{small}$, i.e., $\mathbf{X}_{large}$, when estimating weights. Data similar to the other experiments was generated, albeit only for $k$ = 1 and $\pi$ = 0.0.

## Results

The degree of overlap and average balance in the different scenarios are visualized in S1 and S2 Figs. In Fig 1 the results for the settings common to Experiments 1, 2 and 3, i.e., $n$ = 1500, $MHR_{ATT}$ = 0.8, no censoring, and good overlap ($k$ = 1), are shown. In Scenario (A), all methods except the NAIVE strategy (no weighting) had low biases and similar variances. By increasing the amount of misspecification, the performance of the balancing approach methods CAL-ET and SBW clearly became worse. This was also true for npCBPS, albeit to a lesser extent. Both LASSO and GLM performed well and were quite stable across all scenarios. As can be seen in S1 Table in S1 File, all methods (except NAIVE) in all scenarios resulted in empirical confidence interval coverage very close to or above the nominal 0.95.

Settings with weaker overlap were then considered; in Fig 2 it is clear that variance increases for all weighting methods, as generally does bias. There are, however, considerable differences in the deterioration, with LASSO being the least impacted method. It is noteworthy that, in contrast to Fig 1, GLM did not perform well in the misspecification scenarios, where there was

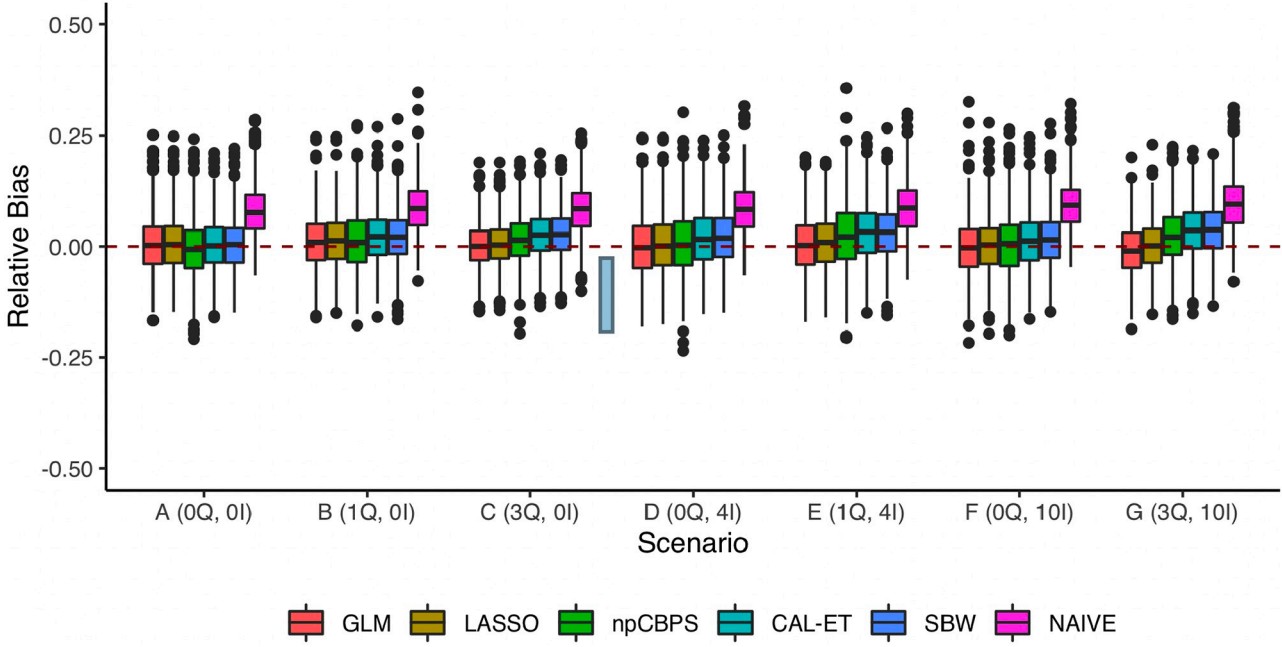

**Fig 1. Relative bias of the $MHR_{ATT}$ estimation methods for different DGPs.** No censoring ($\pi$ = 0) and good overlap ($k$ = 1).

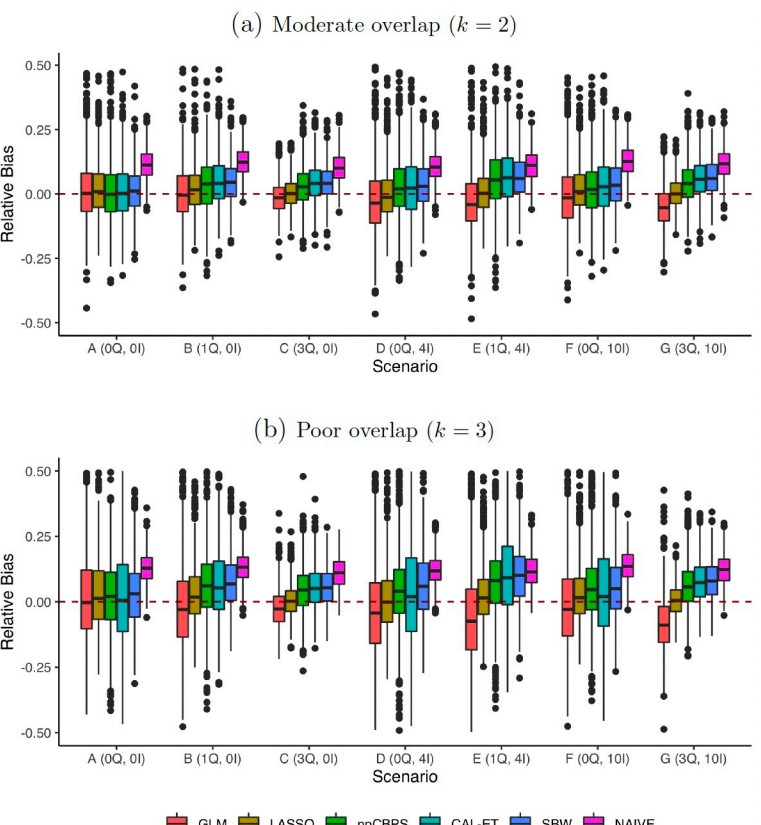

**Fig 2. Experiment 1: Misspecification and overlap.** Relative bias of the $MHR_{ATT}$ estimation methods for varying degrees of overlap and different DGPs. No censoring ($\pi = 0$). Outliers above 0.5 were left out to facilitate visual comparison between the experiments; a complete visualization can be seen in S3 Fig.

poorer overlap. Of the balancing approach methods, SBW resulted in similar or slightly higher bias than CAL-ET, while CAL-ET resulted in higher variance than SBW. npCBPS generally resulted in lower bias than the other balancing approach methods but often higher variance than SBW ($k = 2, 3$) and CAL-ET ($k = 2$). As the overlap became weaker, several methods resulted in poor empirical coverage (S2 and S3 Tables in S1 File). When $k = 3$, LASSO was the only method that resulted in coverage above nominal level in every scenario.

With good overlap but censored data (Fig 3 and S4 Fig; S4-S8 Tables in S1 File) variance increased more than for the uncensored setting, but we also see there was also a tendency for increasing bias as the rate of censoring rose; this started to become notable, while still being low, for higher censoring proportions. The deterioration, however, was not as severe as was seen for the poor overlap uncensored setting. Under the DGPs in this experiment all methods were biased downwards, as can be seen when comparing the low and moderate censoring results. When the censoring rate was 0.4 or higher, GLM and LASSO resulted in empirical coverage of below 0.95 in several scenarios.

The overspecification results show that both CAL-ET and SBW had much lower bias in Scenarios (B)—(G) when constraints were set on $\mathbf{X}_{large}$ (Fig 4; S9 Table in S1 File) instead of $\mathbf{X}_{small}$. This effect was not seen for GLM and LASSO, which already exhibited relatively low bias with $\mathbf{X}_{small}$. All four methods resulted in higher RMSE when using $\mathbf{X}_{large}$. Overall, SBW resulted in the lowest RMSE, closely followed by LASSO. GLM had markedly higher variance

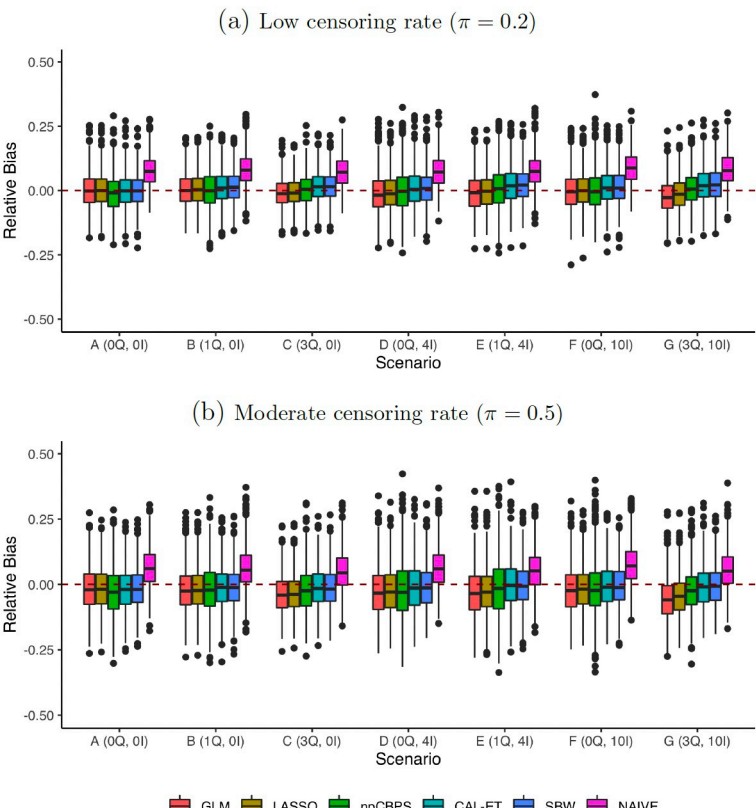

**Fig 3. Experiment 2: Misspecification and censoring.** Relative bias of the $MHR_{ATT}$ estimation methods for varying censoring rates and different DGPs. Good overlap ($k = 1$).

than the other methods in Scenarios (C) and (G). No results are reported for npCBPS since this method was not able to find weights when balance constraints were set on $\mathbf{X}_{large}$.

In S10 Table in S1 File, computational times for the different methods when $n = 1500$ are reported. It took about 11 seconds for LASSO and SBW to find weights using $\mathbf{X}_{small}$, npCBPS

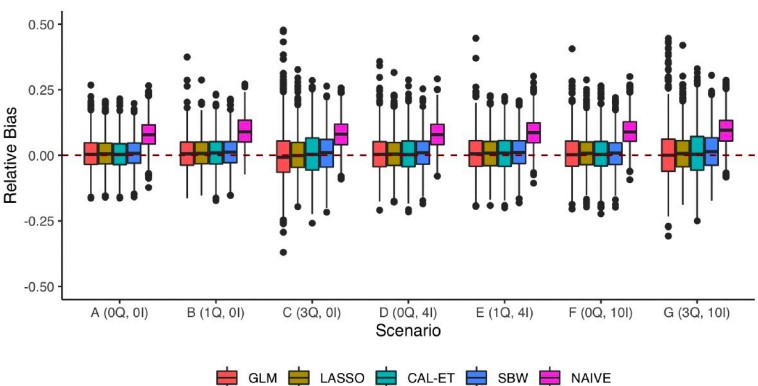

**Fig 4. Experiment 3: Overspecification.** Relative bias of the $MHR_{ATT}$ estimation methods for different DGPs when all models were overspecified. No censoring ($\pi = 0$) and good overlap ($k = 1$). Outliers above 0.5 have been left out to facilitate visual comparison between the experiments; a complete visualization can be seen in S5 Fig.

took around double that time, and GLM and CAL-ET less than 1 second. Finding weights using $\mathbf{X}_{large}$ doubled the time for LASSO and more than tripled the time for SBW.

## Case study

The sample consisted of data from 3400 patients with atrial fibrillation who suffered their first (registered) ischemic stroke in the year 2009, were born in Sweden, and were discharged alive. Of these, 3091 had no missing data and so were used for the analysis. Record linkage register data from the Swedish Stroke Register (Riksstroke), the Swedish Longitudinal Integrated Database for Health Insurance and Labour Market Studies (LISA; managed by Statistics Sweden) and the National Patient Register (NPR; managed by the National Board of Health and Welfare) was used to retrieve information related to the strokes, income, education, and comorbid conditions.

The object of study was the effect of prescribing Warfarin, an oral anti-coagulant, at hospital discharge on the time to either a second ischemic stroke or death within two years of discharge. Thus, patients who survived or had no second stroke within two years after of their hospital discharge had their survival times treated as censored observations. The parameter of interest was the average treatment effect in the population that actually was prescribed Warfarin, hence we estimated $MHR_{ATT}$.

Overall, 1223 (39.57%) patients received a prescription for an anti-coagulant at discharge, while 1868 (60.43%) did not. 1909 of the individuals did not experience an event in the two years after discharge, i.e., 61.76% of the patients were censored. In the analysis, demographic and socioeconomic characteristics, as well as data on risk factors, comorbidities and vital signs on admission were included as plausible confounders (see Table 2 for descriptives and balance (*d*) in the unweighted sample).

As in the simulation study, two different covariate sets were considered for the weighting methods: the untransformed baseline covariates, i.e., $\mathbf{X}_{small}$ (23 terms), and the set which also included quadratic transformations of continuous covariates as well as all two-way interactions, i.e., $\mathbf{X}_{large}$ (187 terms). For combinations of categorical covariates, a minimum sparsity threshold of at least 50 individuals was set.

As can be seen in Table 3, the maximum imbalance when using $\mathbf{X}_{small}$ ranges from 0.37 (LASSO) to 0.78 (NAIVE), clearly exceeding even the not so strict threshold of 0.25. Adequate balance was achieved when both the baseline covariates and transformations of these were included; using $\mathbf{X}_{large}$ resulted in a maximum imbalance of 0.05. Only GLM, LASSO, and SBW were able to deal with the large covariate set, while npCBPS and CAL-ET did not admit solutions to the optimization problem.

For all methods, the Kaplan-Meier survival curves shown in Fig 5 differed significantly between treated and untreated (Log-rank test p- value < 0.0001). Using the survival curves, the absolute reduction in the probability of stroke recurrence or death within two years given the anti-coagulant treatment (absolute risk reduction; ARR) was calculated.

Overlap when estimating the propensity score using $\mathbf{X}_{large}$ was relatively good (S6 Fig). In addition, the proportional hazards assumptions of the Cox models, which were fitted to estimate $MHR_{ATT}$, were tested using scaled Schoenfeld residuals, and all were shown to meet the proportional hazards assumption, except for GLM using $\mathbf{X}_{small}$. As can be seen in Table 4, when adequate balance is achieved, i.e., $\mathbf{X}_{large}$ setting, SBW resulted in point estimates of $MHR_{ATT}$ and ARR that suggest a slightly larger effect of anti-coagulant prescription on time to stroke or death than GLM and LASSO. However, the confidence intervals of all three largely overlapped.

We also estimated $CHR_{ATT}$ by fitting unweighted Cox models which resulted in 0.510 (95% CI: 0.439—0.593) and 0.486 (95% CI: 0.414—0.571) for $\mathbf{X}_{small}$ and $\mathbf{X}_{large}$, respectively.

**Table 2. Characteristics of treated and untreated subjects in the original sample as well as balance between the two groups for each covariate.**

| Covariate | No anti-coagulant (n = 1868) | Anti-coagulant (n = 1223) | Balance |
|---|---|---|---|
| *Demographic and background characteristics* | | | |
| Age | 82.5±8.8 | 76.1±8.8 | **0.73** |
| Female | 1049 (56.2%) | 539 (44.1%) | **0.12** |
| Living alone | 1148 (61.5%) | 517 (42.3%) | **0.19** |
| Living in an institution | 228 (12.2%) | 31 (2.5%) | **0.10** |
| ADL dependency | 253 (13.5%) | 47 (3.8%) | **0.10** |
| *Socioeconomic and educational level* | | | |
| Income | 1680±2012 | 1933±1211 | **0.21** |
| Primary | 1125 (60.2%) | 606 (49.6%) | **0.11** |
| Secondary | 532 (28.5%) | 420 (34.3%) | 0.06 |
| University | 211 (11.3%) | 197 (16.1%) | 0.05 |
| *Level of consciousness at admission* | | | |
| Alert | 1550 (83.0%) | 1139 (93.1%) | **0.10** |
| Drowsy | 273 (14.6%) | 71 (5.8%) | 0.09 |
| Unconscious | 45 (2.4%) | 13 (1.1%) | 0.01 |
| *Comorbid conditions* | | | |
| Diabetes | 347 (18.6%) | 243 (19.9%) | 0.01 |
| Smoking | 140 (7.5%) | 135 (11.0%) | 0.04 |
| Hypertension medication | 1282 (68.6%) | 835 (68.3%) | 0.00 |
| Heart failure | 583 (31.2%) | 296 (24.2%) | 0.07 |
| Ischemic heart disease | 588 (31.5%) | 326 (26.7%) | 0.05 |
| Dementia | 124 (6.6%) | 16 (1.3%) | 0.05 |
| Cancer in last three years | 204 (10.9%) | 116 (9.5%) | 0.01 |
| Valvular disease | 150 (8.0%) | 130 (10.6%) | 0.03 |
| Peripheral arterial disease | 127 (6.8%) | 95 (7.8%) | 0.01 |
| Venous thromboembolism | 68 (3.6%) | 64 (5.2%) | 0.02 |
| Intracerebral hemorrhage (I61) | 31 (1.7%) | 8 (0.7%) | 0.01 |
| Transient ischemic attack (TIA) | 128 (6.9%) | 90 (7.4%) | 0.01 |
| Other major bleeding | 172 (9.2%) | 76 (6.2%) | 0.03 |

*Note*: Continuous covariates reported as mean ± standard deviation. Categorical covariates reported as *n* (%). Covariates with a balance value higher or equal to 0.10 in bold.

**Table 3. Balance distributions, after weighting with GLM, LASSO, npCBPS, CAL-ET, and SBW, as well as the unweighted dataset (NAIVE), when considering balance on all untransformed covariates ($X_{small}$) and all covariates including quadratic terms and interactions (187 terms in total; $X_{large}$).**

| Covariates | Method | Min | 25th Perc. | Median | Mean | 75th Perc. | Max |
|---|---|---|---|---|---|---|---|
| $X_{small}$ | GLM | 0.00 | 0.00 | 0.01 | 0.02 | 0.02 | 0.51 |
| | LASSO | 0.00 | 0.00 | 0.01 | 0.02 | 0.02 | 0.37 |
| | npCBPS | 0.00 | 0.01 | 0.01 | 0.03 | 0.03 | 0.42 |
| | CAL-ET | 0.00 | 0.01 | 0.03 | 0.08 | 0.08 | 0.82 |
| | SBW | 0.00 | 0.00 | 0.00 | 0.01 | 0.01 | 0.48 |
| | NAIVE | 0.00 | 0.01 | 0.05 | 0.09 | 0.10 | 0.73 |
| $X_{large}$ | GLM | 0.00 | 0.00 | 0.00 | 0.01 | 0.01 | 0.05 |
| | LASSO | 0.00 | 0.00 | 0.00 | 0.01 | 0.01 | 0.05 |
| | SBW | 0.00 | 0.00 | 0.00 | 0.00 | 0.00 | 0.05 |

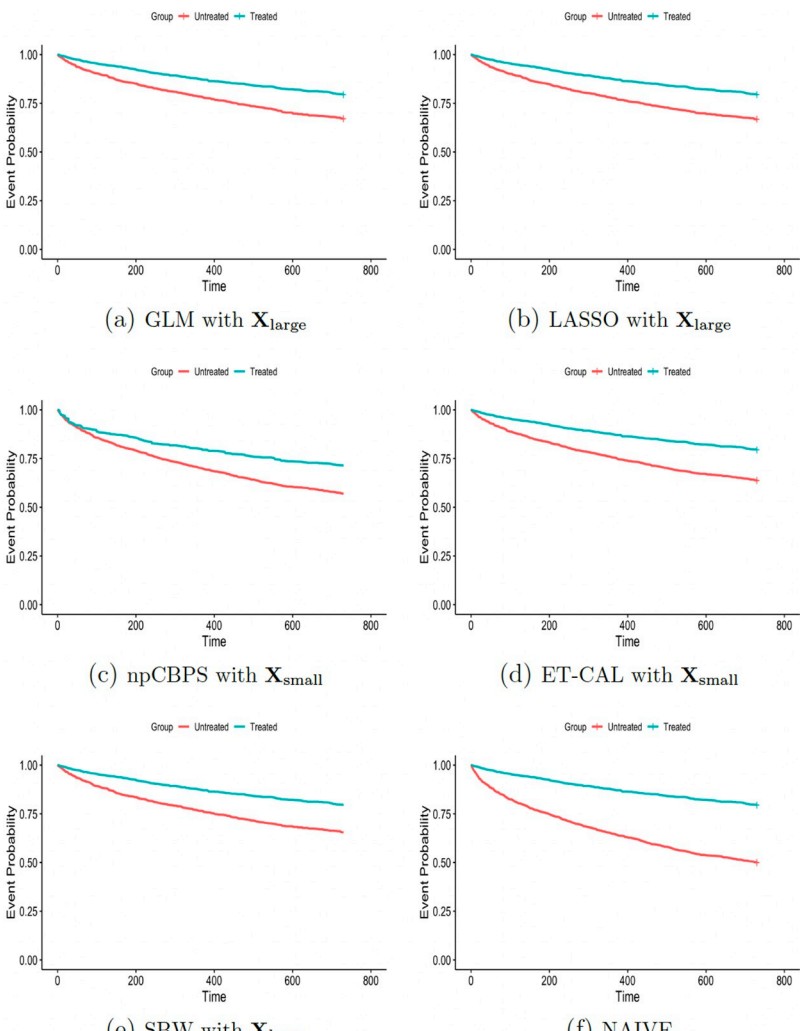

**Fig 5. Kaplan-Meier survival curves for the original sample and the samples weighted by SBW, LASSO, and GLM, respectively.** In this case, survival implies absence of an event (death or ischemic stroke).

**Table 4. Marginal hazard ratio ($MHR_{ATT}$) and absolute risk reduction (ARR) estimates of the anti-coagulant effect on the time to either a second ischemic stroke or death.** CIL and CIU are the lower and upper limits of 95% confidence intervals for $MHR_{ATT}$.

| Covariates | Method | $MHR_{ATT}$ | CIL | CIU | ARR |
|---|---|---|---|---|---|
| $\mathbf{X}_{small}$ | GLM | 0.605 | 0.504 | 0.725 | 0.108 |
| | LASSO | 0.545 | 0.461 | 0.644 | 0.135 |
| | npCBPS | 0.608 | 0.472 | 0.784 | 0.144 |
| | CAL-ET | 0.505 | 0.431 | 0.590 | 0.144 |
| | SBW | 0.513 | 0.438 | 0.599 | 0.152 |
| | NAIVE | 0.326 | 0.284 | 0.374 | 0.294 |
| $\mathbf{X}_{large}$ | GLM | 0.568 | 0.479 | 0.674 | 0.123 |
| | LASSO | 0.561 | 0.476 | 0.661 | 0.127 |
| | SBW | 0.533 | 0.453 | 0.627 | 0.139 |

## Discussion

The results of the simulations revealed that the npCBPS, CAL-ET, and SBW balancing approach methods performed similarly to GLM and LASSO in good overlap scenarios where was no, or low, model misspecification. In bad overlap scenarios with model misspecification, LASSO outperformed the other methods. When models were overspecified, SBW was comparable to LASSO, while CAL-ET and GLM exhibited relatively high variability. npCBPS was unable to deal with the high dimensional dataset resulting from the overspecification. When data was censored, all methods had a downward shift in bias, which in the simulations resulted in poor coverage for the modeling approach methods in scenarios with censoring rates of 40–50%. Due to this downward shift, CAL-ET and SBW performed better than the other methods (in terms of bias and RMSE) in the moderate censoring setting. If a DGP had caused an upward shift in bias this would not have been the case. Recently, Wyss [53] noted that, with censored data, PS estimators of MHR tend to be biased toward CHR, even when censoring is independent, and this is the mechanism behind the results presented in this paper.

The results of the empirical study suggested that anti-coagulant prescription after discharge from hospital following a stroke event has an effect on preventing a second stroke event and increasing the survival of the patient. This is in line with previous results in the literature in which anti-coagulant prescription has been shown to be effective [54–56]. However, since this data falls into the 'good overlap' with 'relatively high censoring' (61.76%) category we suspect that the MHR results are slightly biased.

As shown in this paper, when estimating MHRs with weighting it is important to use methods that target covariate balance, but this does not in itself guarantee good performance in situations with, e.g., poor overlap, high censoring, or misspecified models/balance constraints. However, based on the simulation results we feel confident in recommending LASSO in most settings and the results also support the idea that SBW will often give results similar to LASSO as long as the balance constraints are set on a large enough set of covariates. We also recommend, as in other observational settings, that researchers estimate weights using multiple methods and select the optimal weighting method according to a variety of balancing metrics [57]. It is also worth noting that if, in a poor overlap situation, one is willing to redefine the target population to a subpopulation with good overlap an alternative approach is to use overlap weights [58], which yield exact balance if estimated by logistic regression.

A limitation of the current study is that the results are based on simulations and only reflect the scenarios included in the paper. Although a wide range of scenarios were considered, there are many more that could be of interest, such as scenarios with higher proportion of censoring or more complex variable selection. For the scenarios with higher censoring proportions, weighting related to censoring is a possible inclusion, and there is a need for further investigation regarding how to best estimate MHR under these circumstances. In addition, a sensitivity analysis framework for balancing approach methods has recently been developed [59], but has yet to be explored in the time-to-event setting.

## Supporting information

**S1 Fig. Propensity score overlap based on one replication of simulated data (*n* = 1500) for each scenario and degree of overlap.**
(TIF)

**S2 Fig. Covariate balance averaged over 1000 replicates of simulated data (*n* = 1500) for each scenario and degree of overlap.** The blue and red dashed lines represents average

balance (ASMD or AUD) equal to 0.10 and 0.25, respectively.
(TIF)

**S3 Fig. Experiment 1: Misspecification and overlap.** Relative bias of the $MHR_{ATT}$ estimation methods for varying degrees of overlap and different DGPs. Sample size is $n = 1500$, true $MHR_{ATT} = 0.8$ and no censoring ($\pi = 0$). In Scenario (A) there is no model misspecification. All outliers are included.
(TIF)

**S4 Fig. Experiment 2: Misspecification and censoring.** Relative bias of the $MHR_{ATT}$ estimation methods for varying censoring rates and different DGPs. Sample size is $n = 1500$, true $MHR_{ATT} = 0.8$ and good overlap ($k = 1$). In Scenario (A) there is no model misspecification.
(TIF)

**S5 Fig. Experiment 4: Overspecification.** Relative bias of the $MHR_{ATT}$ estimation methods when all models are overspecified. Sample size is $n = 1500$, true $MHR_{ATT} = 0.8$, no censoring ($\pi = 0$) and good overlap ($k = 1$). All outliers are included.
(TIF)

**S6 Fig. Estimated overlap in the case study data.** The propensity score is estimated using GLM with $\mathbf{X}_{large}$.
(TIF)

**S1 File.**
(ZIP)

## Acknowledgments

The authors are grateful to Associate Professor Maria Josefsson for helpful and constructive comments.

## Author Contributions

**Conceptualization:** Jenny Häggström.

**Data curation:** Marie Eriksson.

**Formal analysis:** Guilherme W. F. Barros.

**Funding acquisition:** Jenny Häggström.

**Investigation:** Guilherme W. F. Barros.

**Methodology:** Guilherme W. F. Barros.

**Project administration:** Jenny Häggström.

**Software:** Guilherme W. F. Barros.

**Supervision:** Marie Eriksson, Jenny Häggström.

**Visualization:** Guilherme W. F. Barros.

**Writing – original draft:** Guilherme W. F. Barros.

**Writing – review & editing:** Guilherme W. F. Barros, Marie Eriksson, Jenny Häggström.

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
