## [Decision Letter · Decision Letter 0]

4 May 2023

PONE-D-23-04363Performance of modeling and balancing approach methods when using weights to estimate treatment effects in observational time-to-event settingsPLOS ONE

Dear Dr. Wang de Faria Barros,

Thank you for submitting your manuscript to PLOS ONE. After careful consideration, we feel that it has merit but does not fully meet PLOS ONE’s publication criteria as it currently stands. Therefore, we invite you to submit a revised version of the manuscript that addresses the points raised during the review process. The authors are requested to make appropriate modifications to this paper as suggested by all reviewers.

We look forward to receiving your revised manuscript.

Kind regards,

Mohamed R. Abonazel, Ph.D.

Academic Editor

PLOS ONE

Journal Requirements:

3. Please ensure that you have specified (1) whether consent was informed and (2) what type you obtained (for instance, written or verbal, and if verbal, how it was documented and witnessed). If your study included minors, state whether you obtained consent from parents or guardians. If the need for consent was waived by the ethics committee, please include this information.

“This work was supported by the Swedish Research Council (Dnr: 2018–01610).

“The authors are grateful to Associate professor Maria Josefsson for helpful and 376 constructive comments. This work was supported by the Swedish Research Council 377 (Dnr: 2018–01610).”

 “This work was supported by the Swedish Research Council (Dnr: 2018–01610).

Additional Editor Comments (if provided):

The authors are requested to make appropriate modifications to this paper as suggested by all reviewers.

Reviewers' comments:

Reviewer's Responses to Questions

**Comments to the Author**

1. Is the manuscript technically sound, and do the data support the conclusions?

Reviewer #1: Partly

Reviewer #2: Yes

Reviewer #3: Partly

2. Has the statistical analysis been performed appropriately and rigorously? 

Reviewer #1: Yes

Reviewer #2: Yes

Reviewer #3: Yes

3. Have the authors made all data underlying the findings in their manuscript fully available?

Reviewer #1: Yes

Reviewer #2: Yes

Reviewer #3: Yes

4. Is the manuscript presented in an intelligible fashion and written in standard English?

Reviewer #1: No

Reviewer #2: No

Reviewer #3: Yes

5. Review Comments to the Author

Reviewer #1: The paper is good. The presentation of the article is also good. The manuscript is technically sound, and do the data support the conclusions. The statistical analysis has been performed appropriately and rigorously. The authors have made all data underlying the findings in their manuscript fully available. The manuscript is not presented in an intelligible fashion and written in standard English.

Reviewer #2: This paper investigates the finite sample properties of different weighting methods when these are used to estimate population level treatment effects using survival data. It is a little research point, however I have some comments to improve the overall look of the paper: the authors should

Review the paper in general linguistically

Adding recent studies that dealt with the same point

Modify the abstract to clarify the purpose of the study, its importance, and the most important findings of their study.

Rewrite the discussion section because it is poor, and clarify the advantages and disadvantages of each method. Explain the reason for preferring one over the other in light of the discussion of previous studies and research.

Reviewer #3: This article compares the modelling vs balancing approach to estimate weights for weighted causal effect estimators of binary exposures and time-to-event outcomes. The article contains sufficient details in the description of the methods, provides a comprehensive simulation study and applies several methods to a real data application. The article is also well written. I only have one major comment, and a few minor comments.

Major comment

1. The sum of the balancing weights is set to 1. In contrast, for estimating the ATT, the sum of the weights based on the propensity score for the control patients will tend to n1, the sample size of the treated patients. This discrepancy would seem to make the comparison between the balancing and modelling approach unfair, as the scaling factor for the weights has an impact on the estimation of the marginal causal hazard ratio in the treated patients. Please set the sum of the balancing weights to be equal to n1 when estimating the ATT.

Minor comments

1. It is worth clarifying on page 8 that the use of the robust variance estimator for estimating the ATT characterized by the marginal hazard ratio was found to lead to conservative inference in simulations only, and that no theoretical justification has been provided. For instance the following article shows that the robust variance estimator can also lead to anti-conservative inference “On Variance of the Treatment Effect in the Treated When Estimated by Inverse Probability Weighting”

2. On page 5, it is worth clarifying that entropy balancing is double robust with respect to a linear outcome regression model and logistic propensity score model. The current setting assumes a log-linear regression model for the hazard, and therefore does not fall in the same setting.

3. It is worth mentioning in the discussion that the overlap weights can also be used as an approach to create exact balance if the user is willing to change the target population to one where there is high overlap in the covariate distributions.

6. PLOS authors have the option to publish the peer review history of their article (what does this mean?). If published, this will include your full peer review and any attached files.

Reviewer #1: No

Reviewer #2: **Yes: **Suzan Abdel-Rahman

Reviewer #3: No

---

## [Author Response · Author response to Decision Letter 0]

18 Jun 2023

The response to all comments can also be seen in the submitted file "Response to Reviewers", but it is reproduced here below:

Dear Editor,

We are grateful for the opportunity to revise our manuscript entitled “Performance of modeling and balancing approach methods when using weights to estimate treatment effects in observational time-to-event settings” in light of the editorial and reviewer comments. Below, we have provided a point-by-point response.

Academic editor's comments

Response:

We have ensured that our manuscript meets PLOS ONE's style requirements.

Response:

All author-generated code has been made available on GitHub. We have provided the following amended data availability statement:

The code used to generate the data in the Monte Carlo simulation is available on the github repository: https://github.com/Wangbarros/Modeling_vs_Balancing_Time_to_Event. The data analyzed in the case study consist of third party data from Riksstroke and according to Swedish legislation (https:// etikprovningsmyndigheten.se/for-forskare/vadsager-lagen/) data cannot be made available for use beyond what has been approved by the ethical review authority. Therefore, the data cannot be made publicly available. Data may be made available from Riksstroke (contact via riksstroke@regionvasterbotten.se) upon reasonable request by researchers who meet the criteria for access to confidential data according to Swedish laws and regulations.

3. Please ensure that you have specified (1) whether consent was informed and (2) what type you obtained (for instance, written or verbal, and if verbal, how it was documented and witnessed). If your study included minors, state whether you obtained consent from parents or guardians. If the need for consent was waived by the ethics committee, please include this information.

Response:

We have added the following sentence to the ethics statement in the “Materials and Methods” section:

"Patients and next of kin are informed about the registration and aim of the Riksstroke-register and their right to decline participation (opt-out consent)."

4. Thank you for stating in your Funding Statement: “This work was supported by the Swedish Research Council (Dnr: 2018–01610). The funders had no role in study design, data collection and analysis, decision to publish, or preparation of the manuscript.” Please provide an amended statement that declares *all* the funding or sources of support (whether external or internal to your organization) received during this study, as detailed online in our guide for authors at http://journals.plos.org/plosone/s/submit- now. Please also include the statement “There was no additional external funding received for this study.” in your updated Funding Statement. Please include your amended Funding Statement within your cover letter. We will change the online submission form on your behalf.

Response:

We have provided the following amended Funding Statement:

"This work was supported by the Swedish Research Council (Dnr: 2018–01610; recipient JH). The funders had no role in study design, data collection and analysis, decision to publish, or preparation of the manuscript. There was no additional external funding received for this study."

5. Thank you for stating the following in the Acknowledgments Section of your manuscript: “The authors are grateful to Associate professor Maria Josefsson for helpful and 376 constructive comments. This work was supported by the Swedish Research Council 377 (Dnr: 2018–01610).”

“This work was supported by the Swedish Research Council (Dnr: 2018–01610). The funders had no role in study design, data collection and analysis, decision to publish, or preparation of the manuscript.” Please include your amended statements within your cover letter; we will change the online submission form on your behalf.

Response:

We have amended the Acknowledgements Section in the manuscript to:

"The authors are grateful to Associate Professor Maria Josefsson for helpful and constructive comments. "

The funding statement has been changed as described in the response to comment 4 above.

Response:

We have now included the following ethics statement in the "Materials and methods" section of the manuscript:

“Statistical method development for fair comparisons of stroke care and outcome was part of the EqualStroke-project, approved by the Ethical Review Board in Umeå (Dnr: 2012-321-31M, 2014-76-32M). Patients and next of kin are informed about the registration and aim of the Riksstroke-register and their right to decline participation (opt-out consent).”

Response:

We have included captions for the Supporting Information files at the end of the manuscript and updated in-text citations.

Reviewer #1

Comment

The paper is good. The presentation of the article is also good. The manuscript is technically sound, and do the data support the conclusions. The statistical analysis has been performed appropriately and rigorously. The authors have made all data underlying the findings in their manuscript fully available. The manuscript is not presented in an intelligible fashion and written in standard English.

Response:

We have engaged a proofreading agency and made changes to the text according to the proofreader's suggestions.

Reviewer #2

Comment

This paper investigates the finite sample properties of different weighting methods when these are used to estimate population level treatment effects using survival data. It is a little research point, however I have some comments to improve the overall look of the paper: the authors should

1. Review the paper in general linguistically

Response:

We have engaged a proofreading agency and made changes to the text according to the proofreader's suggestions.

2. Adding recent studies that dealt with the same point

Response:

We have reviewed recent literature dealing with similar research questions as is the focus of our manuscript and added a few more references and comments on similar work (see citations 57 and 59)

3. Modify the abstract to clarify the purpose of the study, its importance, and the most important findings of their study.

Response:

We have modified the abstract and clarified the purpose, importance and most important findings of the study.

4. Rewrite the discussion section because it is poor, and clarify the advantages and disadvantages of each method. Explain the reason for preferring one over the other in light of the discussion of previous studies and research.

Response:

We have rewritten the discussion.

Reviewer #3

Comment

This article compares the modelling vs balancing approach to estimate weights for weighted causal effect estimators of binary exposures and time-to-event outcomes. The article contains sufficient details in the description of the methods, provides a comprehensive simulation study and applies several methods to a real data application. The article is also well written. I only have one major comment, and a few minor comments.

Major comment

The sum of the balancing weights is set to 1. In contrast, for estimating the ATT, the sum of the weights based on the propensity score for the control patients will tend to n1, the sample size of the treated patients. This discrepancy would seem to make the comparison between the balancing and modelling approach unfair, as the scaling factor for the weights has an impact on the estimation of the marginal causal hazard ratio in the treated patients. Please set the sum of the balancing weights to be equal to n1 when estimating the ATT.

Response:

When we use weights based on the propensity score, for estimating ATT, the sum of the weights for the treated patients is n1 (since each treated patient get weight = 1) and the sum of the weights for the control patient tend to n1 (as the reviewer points out). When we use balancing weights, for estimating ATT, the weights are scaled such that the sum of weights for treated patients is equal to 1 and the sum of the weights for the control patients is also set to 1. Changing the scaling such that both the sum of weights for the treated and control groups would be n1 instead of 1 would not make a difference. If it were the case that the balancing weights for treated summed to n1 then, we agree with the reviewer, that it would be inappropriate to set the sum of the balancing weights to 1.

To clarify this we have added the following sentence in the manuscript when discussing balancing approaches:

“It should be also noted that, for the estimation of ATT weights, all treated individuals receive the same weights, and the sum of the weights for the treated individuals will also be equal to 1.”

Minor comments

1. It is worth clarifying on page 8 that the use of the robust variance estimator for estimating the ATT characterized by the marginal hazard ratio was found to lead to conservative inference in simulations only, and that no theoretical justification has been provided. For instance the following article shows that the robust variance estimator can also lead to anti-conservative inference “On Variance of the Treatment Effect in the Treated When Estimated by Inverse Probability Weighting”

Response:

This has now been clarified and the amended sentience in the manuscript is:

“Although no theoretical justification has been provided, simulation results have indicated that this estimator slightly overestimates the variance, and results in somewhat conservative confidence intervals when estimating MHR_ATE and MHR_ATT with IPTW weights [46]. However, more recently the possibility of anti-conservative inference when estimating ATT with IPTW weights was shown [47]. “

2. On page 5, it is worth clarifying that entropy balancing is double robust with respect to a linear outcome regression model and logistic propensity score model. The current setting assumes a log-linear regression model for the hazard, and therefore does not fall in the same setting.

Response:

The sentence on double robustness of entropy balance has been amended to:

“EB was first proposed as a preprocessing method without considering how this preprocessing would impact any subsequent inference [19], but has since been shown to be doubly robust in settings where the outcome model is linear and the PS model is logistic [34].”

3. It is worth mentioning in the discussion that the overlap weights can also be used as an approach to create exact balance if the user is willing to change the target population to one where there is high overlap in the covariate distributions.

Response:

The following sentence has been included in the Discussion:

"It is also worth noting that if, in a poor overlap situation, one is willing to redefine the target population to a subpopulation with good overlap an alternative approach is to use overlap weights [58], which yield exact balance if estimated by logistic regression."

---

## [Decision Letter · Decision Letter 1]

17 Jul 2023

Performance of modeling and balancing approach methods when using weights to estimate treatment effects in observational time-to-event settings

PONE-D-23-04363R1

Dear Dr. Wang de Faria Barros,

We’re pleased to inform you that your manuscript has been judged scientifically suitable for publication and will be formally accepted for publication once it meets all outstanding technical requirements.

Kind regards,

Mohamed R. Abonazel, Ph.D.

Academic Editor

PLOS ONE

Additional Editor Comments (optional):

Reviewers' comments:

Reviewer's Responses to Questions

**Comments to the Author**

1. If the authors have adequately addressed your comments raised in a previous round of review and you feel that this manuscript is now acceptable for publication, you may indicate that here to bypass the “Comments to the Author” section, enter your conflict of interest statement in the “Confidential to Editor” section, and submit your "Accept" recommendation.

Reviewer #1: All comments have been addressed

Reviewer #3: All comments have been addressed

2. Is the manuscript technically sound, and do the data support the conclusions?

Reviewer #1: Yes

Reviewer #3: Yes

3. Has the statistical analysis been performed appropriately and rigorously? 

Reviewer #1: Yes

Reviewer #3: Yes

4. Have the authors made all data underlying the findings in their manuscript fully available?

Reviewer #1: Yes

Reviewer #3: Yes

5. Is the manuscript presented in an intelligible fashion and written in standard English?

Reviewer #1: Yes

Reviewer #3: Yes

6. Review Comments to the Author

Reviewer #1: The paper is good. The presentation of it is also good. As well as the topic is interesting. I recommend accepting it.

Reviewer #3: (No Response)

7. PLOS authors have the option to publish the peer review history of their article (what does this mean?). If published, this will include your full peer review and any attached files.

Reviewer #1: **Yes: **Dr. Issam Dawoud

Reviewer #3: No

---

## [Editor Report · Acceptance letter]

21 Jul 2023

PONE-D-23-04363R1 

Performance of modeling and balancing approach methods when using weights to estimate treatment effects in observational time-to-event settings 

Dear Dr. Barros:

I'm pleased to inform you that your manuscript has been deemed suitable for publication in PLOS ONE. Congratulations! Your manuscript is now with our production department. 

Kind regards, 

on behalf of

Dr Mohamed R. Abonazel 

Academic Editor

PLOS ONE